# Center-wise Local Image Mixture For Contrastive Representation Learning

## Abstract

Recent advances in unsupervised representation learning have experienced remarkable progress, especially with the achievements of contrastive learning, which regards each image as well its augmentations as a separate class, while does not consider the semantic similarity among images. This paper proposes a new kind of data augmentation, named *Center-wise Local Image Mixture*, to expand the neighborhood space of an image. CLIM encourages both local similarity and global aggregation while pulling similar images. This is achieved by searching local similar samples of an image, and only selecting images that are closer to the corresponding cluster center, which we denote as center-wise local selection. As a result, similar representations are progressively approaching the clusters, while do not break the local similarity. Furthermore, image mixture is used as a smoothing regularization to avoid overconfidence on the selected samples. Besides, we introduce *multi-resolution* augmentation, which enables the representation to be scale invariant. Integrating the two augmentations produces better feature representation on several unsupervised benchmarks. Notably, we reach $75.5\%$ top-1 accuracy with linear evaluation over ResNet-50, and $59.3\%$ top-1 accuracy when fine-tuned with only $1\%$ labels, as well as consistently outperforming supervised pretraining on several downstream transfer tasks.

## 1 Introduction

Learning general representations that can be transferable to different downstream tasks is a key challenge in computer vision. This is usually achieved by fully supervised learning paradigm, *e.g.,* making use of ImageNet labels for pretraining over the past several years. Recently, self-supervised learning has attracted more attention due to its free of human labels. In self-supervised learning, the network aims at exploring the intrinsic distributions of images via a series of predefined pretext tasks (Doersch et al., 2015; Gidaris et al., 2018; Noroozi & Favaro, 2016; Pathak et al., 2016). Among them, instance discrimination (Wu et al., 2018) based methods have achieved remarkable progress (Chen et al., 2020a; He et al., 2020; Grill et al., 2020; Caron et al., 2020). The core idea of instance discrimination is to push away different images, and encourage the representation of different transformations (augmentations) of the same image to be similar. Following this paradigm, self-supervised models are able to generate features that are comparable or even better than those produced by supervised pretraining when evaluated on some downstream tasks, *e.g.,* COCO detection and segmentation (Chen et al., 2020c;b).

In contrastive learning, the positive pairs are simply constrained within different transformations of the same image, *e.g.,* cropping, color distortion, Gaussian blur, rotation, *etc.*. Recent advances have demonstrated that better data augmentations (Chen et al., 2020a) really help to improve the representation robustness. However, contrasting two images that are *de facto* similar in semantic space is not applicable for general representations. It is intuitive to pull semantically similar images for better transferability. DeepCluster (Caron et al., 2018) and Local Aggregation (Zhuang et al., 2019) relax the extreme instance discrimination task via discriminating groups of images instead of an individual image. However, due to the lack of labels, it is inevitable that the positive pairs contain noisy samples, which limits the performance.

In this paper, we target at expanding instance discrimination by exploring local similarities among images. Towards this goal, one need to solve two issues: i) how to select similar images as positive

pairs of an image, and ii) how to incorporate these positive pairs, which inevitably contain noisy assignments, into contrastive learning. We propose a new kind of data augmentation, named *Centerwise Local Image Mixture*, to tackle the above two issues in a robust and efficient way. CLIM consists of two core elements, *i.e.,* a center-wise positive sample selection, as well as a data mixing operation. For positive sample selection, the motivation is that a good representation should be endowed with high intra-class similarity, and we find that although MoCo (He et al., 2020) does not explicitly model invariance to similar images, the intra-class similarity becomes higher as the training process goes. Based on this observation, we explicitly enforce semantically similar images towards the center of clusters, and generate representation with higher intra-class similarity, which we find is beneficial for few shot learning. This is achieved by searching nearest neighbors of an image, and only retaining similar samples that are closer to the corresponding cluster center, which we denote as center-wise local sample selection. As a result, an image is pulled towards the center while do not break the local similarity.

Once similar samples are selected, a direct way is to treat these similar samples as multiple positives for contrastive learning. However, since feature representation in high dimensional space is complex, the returned positive samples inevitably contain noisy assignments, which should not be overconfident. Instead, we rely on data mixing as augmented samples, which can be treated as a smoothing regularization in unsupervised learning. In particular, we apply Cutmix (Yun et al., 2019), a widely used data augmentation in supervised learning, where patches are cut and pasted among the positive pairs to generate new samples. Benefit from the center-wise sample selection, the Cutmix augmentation is only constrained within the local neighborhood of an image, and can be treated as an expansion of current neighborhood space. In this way, similar samples are pulled together in a smoother and robust way, which we find is beneficial for general representation.

Furthermore, we propose *multi-resolution* augmentation, which aims at contrasting the same image (patch) at different resolutions explicitly, to enable the representation to be scale invariant. We argue that although previous operations such as crop and resize introduce multi-resolution implicitly, they do not compare the same patch at different resolutions directly. As comparisons, multi-resolution incorporates scale invariance into contrastive learning, and significantly boosts the performance even based on a strong baseline. The multi-resolution strategy is simple but effective, and can be combined with current data augmentations for further improving performance.

We evaluate the feature representation on several self-supervised learning benchmarks. In particular, on ImageNet linear evaluation protocol, we achieve $75.5\%$ top-1 accuracy with a standard ResNet-50. In few shot setting, when finetuned with only $1\%$ labels, we achieve $59.3\%$ top-1 accuracy, surpassing previous works by a large margin. We also validate its transferring ability on several downstream tasks, and consistently outperform the fully supervised counterparts.

## 2 RELATED WORK

**Unsupervised Representation Learning.** Unsupervised learning aims at exploring the intrinsic distribution of data samples via constructing a series of pretext tasks without human labels. These pretext tasks take many forms and vary in utilizing different properties of images. Among them, one family of methods takes advantage of the spatial properties of images, typical pretext tasks include predicting the relative spatial positions of patches (Doersch et al., 2015; Noroozi & Favaro, 2016), or inferring the missing parts of images by inpainting (Pathak et al., 2016), colorization (Zhang et al., 2016), or rotation prediction (Gidaris et al., 2018). Recent progress in self-supervised learning mainly benefits from instance discrimination, which regards each image (and augmentations of itself) as one class for contrastive learning. The motivation behind these works is the InfoMax principle, which aims at maximizing mutual information (Tian et al., 2019; Wu et al., 2018) across different augmentations of the same image (He et al., 2020; Chen et al., 2020a), (Tian et al., 2019).

**Data Augmentation.** Instance discrimination makes use of several data augmentations, *e.g.,* random cropping, color jittering, horizontal flipping, to define a large view set of vicinities for each image. As has been demonstrated (Chen et al., 2020a; Tian et al., 2020), the effectiveness of instance discrimination methods strongly relies on the type of augmentations. Hoping that the network holds invariance in the local vicinities of each sample. However, current data augmentations are mostly constrained within a single image. An exception is (Shen et al., 2020), where image mixture is used

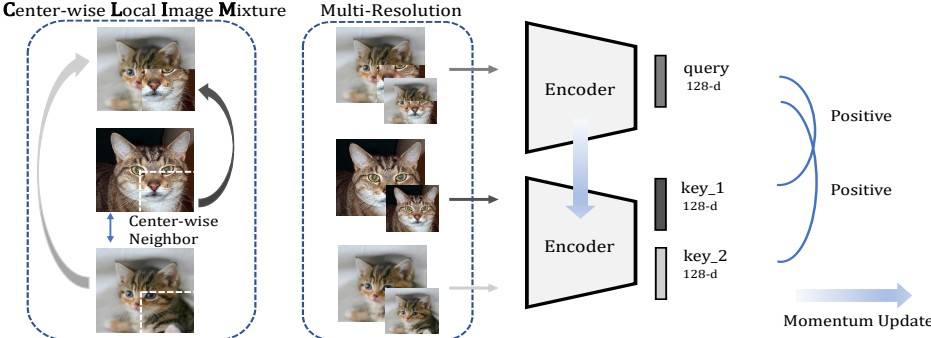

Figure 1: An illustration of the proposed **CLIM** and **multi-resolution** data augmentations.

for flattened contrastive predictions. However, such mixture strategy is conducted among all images, which destroys the local similarity when contrasting mixed samples that are semantic dissimilar.

Beyond self-supervised learning, mixing samples from different images is widely used to help alleviate overfitting in training deep networks. In particular, Mixup (Zhang et al., 2017) combines two samples linearly on pixel level, where the target of the synthetic image was a linear combination of one-hot labels. Following Mixup, there are a few variants (Verma et al., 2018) as well as a recent effort named Cutmix (Yun et al., 2019), which combined Mixup and Cutout (DeVries & Taylor, 2017) by cutting and pasting patches.

## 3 METHOD

In this section, we start by reviewing contrastive learning for unsupervised representation learning. Then we elaborate our proposed CLIM data augmentation, which targets at pulling similar samples via center-wise similar sample selection, followed by a cutmix data augmentation. We also present multi-resolution augmentation that we observe further improves the performance, as well as detailed analysis with recent methods that share similar targets with our method.

### 3.1 CONTRASTIVE LEARNING

Contrastive learning targets at training an encoder to map positive pairs to similar representations while pushing away the negative samples in the embedding space. Given unlabeled training set $X = \{x_1, x_2, ..., x_n\}$. Instance-wise contrastive learning aims to learn an encoder $f_q$ that maps the samples $X$ to embedding space $V = \{v_1, v_2, ..., v_n\}$ by optimizing a contrastive loss. Take the Noise Contrastive Estimator (NCE) (Oord et al., 2018) as an example, the contrastive loss is defined as:

$$\mathcal{L}_{nce}(x_i, x_i') = -\log \frac{\exp(f_q(x_i) \cdot f_k(x_i')/\tau)}{\exp(f_q(x_i) \cdot f_k(x_i')/\tau) + \sum_{j=1}^{K} \exp(f_q(x_i) \cdot f_k(x_j')/\tau))}, \tag{1}$$

where $\tau$ is the temperature parameter, and $x_i'$ and $x_j'$ denote the positive and negative samples of $x_i$, respectively. The encoder $f_k$ can be shared (Chen et al., 2020a; Caron et al., 2020) or momentum update of the encoder $f_q$ (He et al., 2020).

### 3.2 CLIM: CENTER-WISE LOCAL IMAGE MIXTURE

In contrastive learning, each sample as well as its augmentations is treated as a separate class, while all other samples are regarded as negative examples and pushed away. In principle, semantically similar samples should have similar feature representation in the embedding space, while current contrastive strategies do not consider the semantic similarities among different samples, and only choose different views of the same sample as positive pairs. To solve this issue, we propose a new kind of data augmentation, termed as Center-wise Local Image Mixture, which pulls samples that

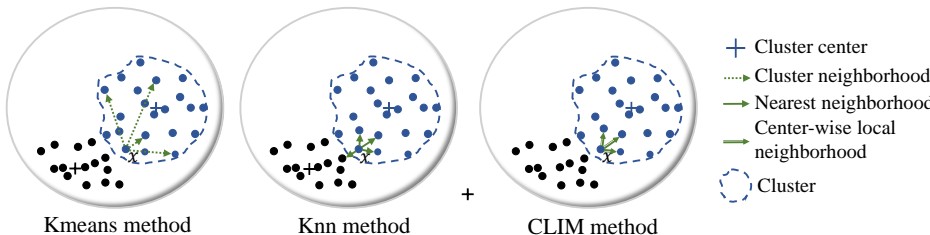

Figure 2: Comparison of three positive sample selection strategies, *i.e.,* k-means, knn, and the proposed center-wise local sample selection.

are semantically similar in an efficient and robust way. The proposed CLIM augmentation consists of two elements, *i.e.,* center-wise local similar sample selection, and a cutmix data augmentation, which would be described in details in the following.

### 3.2.1 CENTER-WISE LOCAL POSITIVE SAMPLE SELECTION

As noted by (Wang & Isola, 2020), a good representation should satisfy both alignment and uniformity, which encourages similar images to have similar representation in the embedding space, and meanwhile, semantically similar features are well-clustered. Towards this goal, we propose a positive sample selection strategy that considers both local similarity and global aggregation. This is achieved by searching similar samples within a cluster that the anchor sample belongs to, and only retaining samples that are closer to the corresponding cluster center. We denote it as center-wise local selection as these samples are picked out towards the cluster center among the local neighborhood of an image. In this way, similar samples are progressively pulled to the predefined cluster centers, while do not break the local similarity.

Specifically, given a set of unlabeled images $X = \{x_1, x_2, ..., x_n\}$ and the corresponding embedding $V = \{v_1, v_2, ..., v_n\}$ with encoder $f_\theta$, where $v_i = f_\theta(x_i)$. We cluster the representations $V$ using a standard k-means algorithm, and obtain $m$ centers $C = \{c_1, c_2, ..., c_m\}$. Given an anchor $x_i$ with its assigned cluster $c(x_i) \in C$, denote the sample set that belongs to $c(x_i)$ as $\Omega_1 = \{x|c(x) = c(x_i)\}$. We search the $k$ nearest neighbors of $x_i$ over the entire space with L2 distance, obtaining sample set $\Omega_2 = \{x_{i1}, ..., x_{ik}\}$. The positive samples are selected based on the following rule:

$$\Omega_p = \{x|d(f_\theta(x), v_{c(x_i)}) \le d(f_\theta(x_i), v_{c(x_i)}), x \in \Omega_1 \cap \Omega_2\}, \tag{2}$$

where $d(\cdot, \cdot)$ denotes the L2 distance of two samples, and $v_{c(x_i)}$ denotes the feature representation of the corresponding cluster center, respectively. In this way, the samples are aggregated towards the predefined clusters, and meanwhile maintaining the local similarity.

Our method combines the advantages of cluster and nearest neighbor methods. An illustration comparing the three methods is shown in Fig. 2. Cluster-based method regards all samples that belong to the same center as positive pairs, which breaks the local similarity among samples especially when the anchor is around the boundary. While nearest neighbor-based method independently pulling samples of an anchor, and does not encourage the well-clustered goal. As a result, the embedding space is not highly concentrated among multiple similar anchors. As comparisons, by center-wise sample selection, similar samples are progressively pulled to the predefined center as well as considering the local similarity. In the experimental section, we would compare the performance of the three methods, and validate the superior performance of our proposed selection strategy.

### 3.2.2 CUTMIX DATA AUGMENTATION

Once we obtain the positive samples of an anchor, one direct way is to treat these samples similar as the augmented ones for contrastive learning. However, similarity computation in high dimensional space inevitably contains noisy samples, which should not be overconfident for contrasting. To solve this issue, we make use of data mixture strategy, which aims at mixing patches from two different images as augmented samples for contrasting. Data mixing is widely used in supervised learning as label smoothing regularization. The highlight is that without image level labels, we are not able to assign new labels to the augmented samples. Instead, we only mixing samples that are similar

in representation, and the mixed samples can be treated as an augmented version of the anchor. In this way, these mixed samples, as well as traditional data augmentations, can be pulled together in contrastive learning. Specifically, given a positive pair $(x_i, \tilde{x}_i)$, we conduct data mixing as follows:

$$x_{mix} = \mathbf{M} \odot x_i + (\mathbf{1} - \mathbf{M}) \odot \tilde{x}_i, \tag{3}$$

where $\mathbf{M} \in \{0, 1\}^{W \times H}$ denotes a binary mask indicating the mixed rectangle region of an image, *i.e.,* where to cutout the region in $x_i$ and replaced with a randomly selected patch from $\tilde{x}_i$, and $W, H$ denotes the wide and height of an image, respectively. $\mathbf{1}$ is a binary mask filled with ones, and $\odot$ is the element-wise multiplication operation. For mask $\mathbf{M}$ generation, we follow the setting in (Yun et al., 2019). For the mixed sample $x_{mix}$, the positive sample can be either $x_i$ or $\tilde{x}_i$, and we reformulate the contrastive learning as combing two NCE loss:

$$\mathcal{L}_{mix}(x_i, \tilde{x}_i) = \lambda \cdot \mathcal{L}_{nce}(x_{mix}, x_i) + (1 - \lambda) \cdot \mathcal{L}_{nce}(x_{mix}, \tilde{x}_i). \tag{4}$$

Where the combination ratio $\lambda$ is sampled from beta distribution Beta$(\alpha, \alpha)$ with parameter $\alpha$. The proposed data mixing augmentation can be seamlessly incorporated into current contrastive learning. The advantages are twofold: first, mixed samples help to expand the neighborhood space of current anchor sample for better representation; second, minimizing the two terms simultaneously can help to maximize the mutual information between $x_i$ and $\tilde{x}_i$ in a soft manner and perform as smoothing regularization on the prediction for selected positive samples.

### 3.3 Multi-resolution Data Augmentation

Data augmentation plays a key role in current contrastive learning, among them crop augmentation is one of the most effective way (Chen et al., 2020a). In a typical crop augmentation, a sample $x$ with size $H \times W$ is randomly cropped with ratio $\sigma$, and resized to $K_{train} \times K_{train}$ as augmented samples, where $K_{train} \times K_{train}$ denotes the input resolution for model training. Hence the scaling factor w.r.t. sample $x$ can be described as:

$$s = \frac{1}{\sigma} \cdot \frac{K_{train}}{\sqrt{H \times W}}. \tag{5}$$

For crop augmentation, the parameter $K_{train}$ is fixed, and the crop ratio $\sigma$ is randomly selected among positive pairs. As a result, different crop augmentations usually contain different contents, which can be regarded as modeling occlusion invariance to some extent, where each crop sees one view of an image. In this section, we propose a simple but effective data augmentation strategy, named multi-resolution augmentation, which enables the representation to be scale invariant of an example. The highlight is that it is better for contrasting positive pairs with the same content but different resolutions. Specifically, for each positive we keep the crop ratio $\sigma$ fixed, and adjust $K_{train}$ to different resolutions for contrastive loss. An illustration is shown in Fig. 1 .Using multi-resolution, the objective function can be generalized as:

$$\mathcal{L}_{mr} = \sum_{r,r' \in \{r_1, ..., r_n\}} \mathcal{L}_{mix}(x_i^r, \tilde{x}_i^{r'}), \tag{6}$$

where $\{r_1, ..., r_n\}$ indicates the resolution set. In this way, the encoder would be encouraged to discriminate the positive samples with different resolutions from a series of negative keys, which will maximize the mutual information between inputs with different resolutions and discard redundant information brought by resolutions.

**Relation with Multi-crop Augmentation.** There exist recent works that aim at improving crop augmentations, including multi-crop (Caron et al., 2020) and jigsaw-crop (Misra & Maaten, 2020). However, both methods target at reducing crop ratio $\sigma$ in Eq.5 and resolution $K_{train}$ simultaneously to bridge different parts of an object, and do not explicitly model scale invariance. As comparisons, our proposed multi-resolution strategy fixes the crop ratio to explicitly model scale invariance. In the experimental section, we would compare these two augmentations to validate the difference.

Table 1: Top-1 accuracies under linear evaluation on ImageNet, using ResNet-50 as encoder

| Method | Accuracy (%) |
|---|---|
| Supervised | 76.5 |
| Colorization (Zhang et al., 2016) | 39.6 |
| Jigsaw (Noroozi & Favaro, 2016) | 45.7 |
| NPID (Wu et al., 2018) | 54.0 |
| LA (Zhuang et al., 2019) | 58.8 |
| MoCo (He et al., 2020) | 60.6 |
| SeLa (YM. et al., 2020) | 61.5 |
| PIRL (Misra & Maaten, 2020) | 63.6 |
| CPCv2 (Hénaff et al., 2019) | 63.8 |
| PCL (Li et al., 2020) | 65.9 |
| SimCLR (Chen et al., 2020a) | 70.0 |
| MoCo v2 (Chen et al., 2020c) | 71.1 |
| SimCLRv2 (Chen et al., 2020b) | 71.7 |
| InfoMin (Tian et al., 2020) | 73.0 |
| BYOL (Grill et al., 2020) | 74.3 |
| SwAV (Caron et al., 2020) | 75.3 |
| CLIM | **75.5** |

Table 2: Semi-supervised learning with few shot ImageNet labels, using ResNet-50 as encoder (averaged by 5 trials)

| Method | Top-1 / Top-5 | | | |
|---|---|---|---|---|
| | 1% labels | | 10% labels | |
| Supervised | 25.4 | 48.4 | 56.4 | 56.4 |
| PIRL | 30.7 | 57.2 | 60.4 | 83.8 |
| SimCLR | 48.3 | 75.5 | 65.6 | 87.8 |
| MoCo v2 | 52.4 | 78.4 | 65.3 | 86.6 |
| BYOL | 53.2 | 78.4 | 68.8 | 89.0 |
| SwAV | 53.9 | 78.5 | **70.2** | **89.9** |
| SimCLRv2 | 57.9 | **82.5** | 68.4 | 89.2 |
| CLIM | **59.3** | 81.6 | 70.0 | 89.3 |

Table 3: Transfer learning on VOC object detection (averaged by 5 trials).

| Method | Accuracy (%) | |
|---|---|---|
| | $AP_{50}$ | $AP_{75}$ |
| Supervised | 81.4 | 58.8 |
| MoCo v2 | 82.5 | 64.0 |
| SwAV | 82.6 | - |
| CLIM | **82.8** | **64.5** |

**Relation with Fix-Res.** The proposed multi-resolution augmentation is reminiscent of recent work FixRes (Touvron et al., 2019), which also explores resolution issue of better representation, but they are different in both motivation and goal. FixRes is based on the observation that data augmentations induce a significant discrepancy between the size of the objects seen by the classifier at train and test time, and employs different train and test resolutions to fix the train-test resolution discrepancy. The goal is to require less scale invariance for the neural net in FixRes. While our multi-resolution augmentation aims to model the scale invariance explicitly, which is not carefully considered in previous self-supervised learning.

## 4 EXPERIMENTAL RESULTS

In this section, we assess our pretrained feature representation on several unsupervised benchmarks. We evaluate it on ImageNet under linear evaluation and semi-supervised settings. Then we transfer the learned features to different downstream tasks. We also analyze the performance of our representation with detailed ablation studies. For brief expression, except for the ablation study, we denote our method as CLIM, which includes two kinds of data augmentations.

### 4.1 LINEAR EVALUATION ON IMAGENET

The feature representation is trained based on ImageNet 2012 (Russakovsky et al., 2015), using a standard ResNet-50 structure as backbone. We follow the setting in MoCo v2 (Chen et al., 2020c), and the training details are listed in Appendix A. We first evaluate our features by training a linear classifier on top of the frozen representation, following a common protocol in (He et al., 2020; Tian et al., 2019). For linear classifier, the learning rate is initialized as 30 and decayed by 0.1 after 60, 80 epochs, respectively. Table 1 shows the top-1 accuracies with center crop evaluation. Our method achieves an accuracy of 75.5%, surpassing MoCo v2 baseline (71.1%) by 4.4%, and nearly approaching the supervised learning baseline (76.5%).

Table 4: Transfer learning on COCO detection and instance segmentation (averaged by 5 trials)

| Method | Mask R-CNN,R50-FPN,Det | | | | | | Mask R-CNN,R50-FPN,InsSeg | | | | | |
|---|---|---|---|---|---|---|---|---|---|---|---|---|
| | $1\times$ schedule | | | $2\times$ schedule | | | $1\times$ schedule | | | $2\times$ schedule | | |
| | $AP^{bb}$ | $AP^{bb}_{50}$ | $AP^{bb}_{75}$ | $AP^{bb}$ | $AP^{bb}_{50}$ | $AP^{bb}_{75}$ | $AP^{mk}$ | $AP^{mk}_{50}$ | $AP^{mk}_{75}$ | $AP^{mk}$ | $AP^{mk}_{50}$ | $AP^{mk}_{75}$ |
| Supervised | 38.9 | 59.6 | 42.0 | 40.6 | 61.3 | 44.4 | 35.4 | 56.5 | 38.1 | 36.8 | 58.1 | 39.5 |
| MoCo v2 | 39.2 | 59.9 | 42.7 | 41.5 | 62.2 | 45.3 | 35.7 | 56.8 | 38.1 | 37.5 | 59.1 | 40.1 |
| CLIM | **39.5** | **60.0** | **43.3** | **41.8** | **62.3** | **45.7** | **35.8** | **57.0** | **38.6** | **37.7** | **59.4** | **40.5** |

## 4.2 Semi-supervised training on ImageNet

We also evaluate our method by fine-tuning the pretrained model with a small subset of labels, following the semi-supervised settings in (Grill et al., 2020; Kornblith et al., 2019; Chen et al., 2020a; Caron et al., 2020). For fair comparisons, we use the same fixed $1\%$ and $10\%$ splits of training data as in (Chen et al., 2020a), and fine-tune all layers using SGD optimizer with momentum of 0.9, and learning rate of 0.0001 for backbone, 10 for the newly initialized fc layer. The fine-tune epochs is set as 60, and the learning rate is decayed by 0.1 after every 20 epochs. During training, only random cropping and flipping data augmentations are used for fair comparison. The results are reported in Table 2. CLIM achieves $59.3\%$ top-1 accuracy with only $1\%$ labels, and $70.0\%$ with $10\%$ labels. The performance gains are larger with $1\%$ labels, *e.g.,* $6.1\%$ higher than BYOL, and $5.4\%$ better than SwAV, which demonstrates that the proposed feature representation is mainly suitable for extremely few shot learning. Note that SimCLR v2 makes use of other tricks like more MLP layers for better performance, while our method simply adds one fc layer, and still achieves better performance under both settings.

## 4.3 Downstream tasks

We also evaluate our feature representation on several downstream tasks, including object detection and instance segmentation, to evaluate the transferability of the learned features. For fair comparison, all experiments follow MoCo settings.

**PASCAL VOC Object Detection.** Following the evaluation protocol in (He et al., 2020), we use Faster R-CNN (Ren et al., 2015) with R50-C4 as backbone. We fine-tune all layers on the trainval set of VOC07+12 for $2\times$ schedule and evaluate on the test set of VOC2007. We report the performances under the metric of AP50 and AP75. As shown in Table 3, on PASCAL VOC, CLIM achieves $82.8\%$ and $64.5\%$ mAP under AP50 and AP75 metric, which is 1.4 points and 5.7 points higher than the fully supervised counterparts, and is slightly better than the results of MoCo v2.

**COCO Object Detection and Instance Segmentation.** We also evaluate the representation learned on a large scale COCO dataset. Following (He et al., 2020), we choose Mask R-CNN with FPN as backbone, and fine-tune all the layers on the train set and evaluate on the val set of COCO2017. In Table 4, we report results under both $1\times$ and $2\times$ schedules. We show that CLIM consistently outperforms the supervised pretrained model and MoCo v2. Under 2X schedule, we achieve $41.8\%$ and $37.7\%$ detection and segmentation accuracies, respectively, which is 1.2 points and 1.1 points better than the supervised couterparts, and also slightly better than the highly optimized MoCo v2.

**LVIS Long Tailed Instance Segmentation.** Different from VOC and COCO where the number of training samples is comparable, LVIS is a long-tailed dataset, which contains more than 1200 categories, among them some categories only have less than ten instances. The main challenge is to learn accurate few shot models for classes among the tail of the class distribution, for which little data is available. We evaluate our features on this long-tailed dataset to validate how the unsupervised representation boosts the performance. Similarly, we fine-tune the model (Mask R-CNN, R50-FPN) on the train set and evaluate on the val set of Lvis v0.5. Table 5 shows the result under $2\times$ schedule. CLIM outperforms the supervised pretrained model by a large margin and is slightly better than MoCo v2. We claim that it is mainly to the proposed data mixing data augmentation, which is able to learn generalized representations even with extremely few labeled data.

Table 5: Transfer learning on LVIS long-tailed instance segmentation (averaged by 5 trials)

| Method | Object Det | | | Instance Seg | | |
|---|---|---|---|---|---|---|
| | $AP^{bb}$ | $AP^{bb}_{50}$ | $AP^{bb}_{75}$ | $AP^{bb}$ | $AP^{mk}_{50}$ | $AP^{mk}_{75}$ |
| Supervised | 24.1 | 39.4 | 25.0 | 24.2 | 37.8 | 25.1 |
| MoCo v2 | 25.1 | 40.4 | 26.1 | 25.3 | 38.4 | 27.0 |
| CLIM | **25.5** | **41.2** | **26.7** | **25.6** | **39.5** | **27.5** |

Table 6: Impact of different sample selection

| Strategy | Accuracy (%) | |
|---|---|---|
| | no mixing | +cutmix |
| MoCo v2 | 67.5 | - |
| Random | 62.3 | 67.1 |
| KNN | 68.3 | 69.5 |
| K-means | 68.0 | 69.2 |
| KNN ∩ K-means | 68.5 | 69.6 |
| Center-wise | **69.3** | **70.1** |

Table 7: Impact of different multiple resolutions

| Method | Resolution | Accuracy (%) |
|---|---|---|
| Multi-Crop | $2\times224 + 2\times96$ | 69.7 |
| Multi-Reso | $r, r' \in \{224, 96\}$ | 70.4 |
| | $r, r' \in \{224, 128\}$ | 71.7 |
| | $r, r' \in \{224, 160\}$ | **72.3** |
| | $r, r' \in \{224, 224\}$ | 71.4 |

## 4.4 ABLATION STUDY

In this section, we present ablation studies to better understand how each component affects the performance. Detailed comparisons include 1) positive sample selection, 2) cutmix data augmentation, and 3) multi-resolution augmentation. Unless specified, we train the model for 200 epochs over the ImageNet-1000 and report the top-1 classification accuracy under linear evaluation protocol.

**Positive Sample Selection.** We first analyze the advantages of our proposed center-wise local sample selection strategy. The compared sample selection alternatives include:

• Random selection: Randomly select a sample from all unlabeled data.

• KNN selection: Use k-nearest neighbors to build the correlation map among samples, and randomly select a sample from the Top-$k$ ($k = 10$) nearest neighbors as positive samples.

• K-means selection: Use k-means clustering algorithm to obtain $k$ cluster centers, and randomly select a sample from the corresponding cluster as positive samples.

• KNN ∩ K-means selection: Use K-means clustering algorithm to obtain $k$ cluster centers, and randomly select nearest neighbor within the cluster as positive samples.

The results are shown in the second column of Table 6. In order the inspect the influence of sample selection, we do not conduct cutmix augmentation, and these positive samples are simply pulled via a standard contrastive loss. It can be shown that comparing with the MoCo baseline, both KNN and cluster-based sample selection boost the performance, and notably, simply selecting the union of knn and k-means achieves 68.5% accuracy, which is comparable with result that directly using knn. Since for samples not lie around the boundary, it equals to knn, and does not encourage intra-class compactness. As comparison, our proposed center-wise selection strategy outperforms all the above selection methods.

**Cutmix Data Augmentation.** Data mixing helps to expand the neighborhood space of the target sample, and acts as smoothing regularization for the prediction. As shown in the third column of Table 6, cutmix augmentation consistently improve the performance, comparing with directly pulling similar samples in contrastive loss, and achieve 70.1% accuracy with only 200 training epochs. Notably, with randomly selected positive samples, cutmix operation even obtains 67.1% accuracy, slightly lower than the MoCo baseline, while significantly better than no mixing with only 62.3% accuracy. This can be attributed to the smoothing regularization of cutmix, which is able to alleviate the effect of noisy samples and update model in a more robust way.

**Multiple Resolution.** Based on CLIM, we further add multi-resolution data augmentation to validate its effectiveness. The results of introducing different resolutions are shown in Table 7. Using multiple resolutions setting with $r, r' \in \{224, 160\}$, our method achieves an accuracy of 72.3% with

only 200 epochs, which surpasses the baseline of MoCo by $4.8\%$, and even much better than the results of MoCo with 800 epochs ($71.1\%$).

We also compare our multi-resolution augmentation with multi-crop augmentation proposed in (Caron et al., 2020). $2 \times 224 + 2 \times 96$ denotes using two $224 \times 224$ crops with crop-scale $\sigma \sim U(0.2, 1.0)$ and two $96 \times 96$ crops with $\sigma \sim U(0.05, 0.14)$, referring to (Caron et al., 2020). The main difference is that, the multi-crop strategy targets at capturing relationship between local and global information, while our proposed multiple resolution target at enabling the encoder with scale invariance. We find that multi-crop slightly deteriorates the performance of CLIM ($70.1\%$ versus $69.7\%$), partially because data mixing behaves like image cropping augmentation, and shares similarity with multi-crop strategy.

## 5 CONCLUSION

In this work, we proposed CLIM data augmentation, to efficiently pull semantically similar samples for better representation in contrastive learning. The main contributions of CLIM consist of two elements, center-wise positive sample selection, which considers both local similarity and global aggregation property. In such way, similar samples are progressively aggregated to a series of predefined clusters, while not breaking the local similarity; and data mixing augmentation, which expands the neighborhood space of an example by mixing two images, and acts as a smoothing regularization for contrastive loss. Furthermore, we present a simple but effective multi-resolution augmentation, which explicitly model scale invariance to further improve the representation. Experiments evaluated on several unsupervised benchmarks demonstrate the effectiveness of our method.

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

# A   IMPLEMENTATION DETAILS

## A.1   IMPLEMENTATION DETAILS FOR CONTRASTIVE PRETRAINING

**Architecture and Optimization**. We follow the setting in MoCo v2 (Chen et al., 2020c), which relies on two encoders, one for training and the other one for momentum update ($m = 0.999$) to store negative keys. Following SimCLR (Chen et al., 2020a), we replace the fc head with a 2-layer MLP to project the output of the final pooling layer to 128-d. We use SGD as optimizer, with weight decay setting as 0.0001 and the momentum as 0.9. We use a mini-batch size of 512 on 16 V100 GPUs with a cosine learning rate schedule decayed from 0.06. We train the model for 1200 epochs, as we introducing data mixing augmentation, and usually requires more epochs for better performance as in supervised learning (Yun et al., 2019). [1]

**Image Augmentations**. We combine the proposed augmentations with previous widely used basic augmentation strategies, following the settings in (Chen et al., 2020a; He et al., 2020). The basic augmentations are listed below, as well as the corresponding parameters.

- **RandomResizedCrop**: A crop of random size (from 0.2 to 1.0) of the original size and a random aspect ratio (from 3/4 to 4/3) of the original aspect ratio is made.
- **RandomFlip**: Randomly horizontally flip the image with a probability of 0.5.
- **ColorJitter**: Randomly change the brightness, contrast and saturation of an image.
- **RandomGrayscale**: Randomly convert RGB image to grayscale with a probability of 0.2.
- **RandomGaussianBlur**: Randomly blur the image with a probability of 0.5. The radius is randomly sampled from 0.1 to 2.0.

## A.2   DETAILS OF POSITIVE SAMPLE SELECTION

We implement k-means and knn by faiss (Johnson et al., 2019). For efficiency, we perform clustering and knn computation every 5 epochs, since each iteration can be finished within minutes, the extra computation cost is marginal comparing with the budget for model training. The number of clusters is set as $10K$, and we select the top 40 nearest neighbors in knn. In order to balance the contribution of each image, we randomly select 10 positive samples for the following cutmix augmentations. For situations where there remained no more than 10 examples (*e.g.,* the anchor is already around the cluster center), we simply select the most nearest samples among the remained top-40 knn samples.

# B   MORE ABLATION STUDIES

This section gives more detailed analysis w.r.t. some hyperparameters. Unless specified, we train the model for 200 epochs over the ImageNet-1000 and report the top-1 classification accuracy under linear evaluation protocol.

Table 8: Impact of the number of clusters $m$ and $k$ of knn

| Number of Clusters ($m$) | 5000 | | | **10000** | | | 20000 | | |
|---|---|---|---|---|---|---|---|---|---|
| knn ($k$) | 20 | 40 | 60 | 20 | **40** | 60 | 20 | 40 | 60 |
| Accuracy (%) | 69.1 | 69.5 | 69.4 | 70.0 | **70.1** | 69.7 | 69.6 | 69.9 | 69.5 |

**The number of Clusters $m$ and the $k$ in Knn**. Here we inspect the impact of the number of clusters $m$ in k-means and the $k$ in knn to analyze their effect on the performance. In order to ensure local similarity, we restrict the nearest neighbors within a range from 20 to 60. The results for different clusters and top-k neighbors are shown in Table 8. We observe that CLIM consistently improves

---

[1]It is hard for fair comparison w.r.t. training epochs, since different methods make use of different epochs and batchsize. *e.g.,* BYOL and SimCLR report results on 1000 epochs, while MoCo and SwaV are 800 epochs. We empirically find that for MoCo, more training epochs do not improve the performance further.

the performance comparing the baseline Moco $67.5\%$, and is relatively robust to different $m$ and $k$. Notably, the best performance is achieved when $m = 10000$, $k = 40$.

**Hyperparameters $\alpha$ in Cutmix**. The combination $\lambda$ in cutmix is sampled from the beta distribution Beta$(\alpha, \alpha)$, where $\alpha$ plays an important role in data mixing augmentation, which controls the strength of interpolation between the anchor and its positive pair. Here we inspect how different $\alpha \in \{1, 1.5, 2, 2.5\}$ affect the representation. As shown in Table 9. We find that the performance is relatively robust to different $\alpha$, and the best performance is achieved when $\alpha$ is set as 2.

Table 9: Impact of $\alpha$ in cutmix

| $\alpha$ | 1.0 | 1.5 | **2.0** | 2.5 |
|---|---|---|---|---|
| Accuracy (%) | 69.7 | 69.9 | **70.1** | 69.8 |

**Ablation study on mixing strategies**. Our method targets at generating new samples that expanding the neighborhood of an anchor. Here we compare performance of using mixup data augmentation, a widely used method in supervised settings. We try different choices of beta distribution for Mixup (Zhang et al., 2017) and choose the best one ($\alpha = 0.2$) for comparison. Table 10 shows that Cutmix performs better than Mixup, partially because mixup destroys the real pixel distribution (destroys the naturality of pixels).

Table 10: Ablation study on the mixing methods

| Method | Accuracy (%) |
|---|---|
| Mixup | 69.5 |
| Cutmix | 70.1 |

**Extra ablation experiments for longer training schedule**. We compare the improvements brought by different components of our proposed method for longer training schedule (800 epochs). Table 11 shows the top-1 accuracies under linear evaluation protocol. Our method consistently outperforms the MoCo v2 baseline, which demonstrates the effectiveness of our proposed method.

Table 11: Ablation study on the longer training schedule

| Method | Accuracy (%) |
|---|---|
| MoCo v2 | 71.1 |
| Center-wise + cutmix | 73.7 |
| Center-wise + cutmix + Multi-reso | 75.2 |

## C   MORE EXPERIMENTAL RESULTS

**Visualization of Feature Representation**. We visualize the feature space to better understand how CLIM augmentation pulls similar samples. Specifically, we randomly choose 10 classes from the validation set and provide the *t-sne* visualization of feature representation generated by CLIM, supervised training and MoCo v2. As shown in Fig. 3, the same color denotes features with the same label. It can be shown that CLIM takes on higher aggregation property comparing with MoCo, and the fully supervised learned representation reveals the highest aggregation due to it makes use of image labels. Furthermore, we compute the intra-class similarity as the average cosine distance among all intra-class pairwise samples, and report the average similarity across 1000 classes, as shown in Table 12, CLIM achieves an intra-class similarity of $0.65$, which is much higher than that in MoCo v2 with similarity of only $0.58$. As comparison, we also list the result of supervised learning, with a similarity metric of $0.75$.

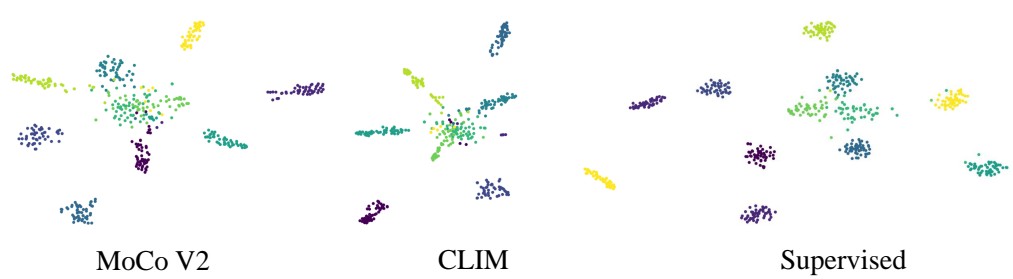

MoCo V2           CLIM           Supervised

Figure 3: *t-sne* visualization of representation learned by MoCo, CLIM and supervised learning.

Table 12: Intra-class similarity for different models

| Method | Intra-class Similarity |
|--------|------------------------|
| Supervised | 0.75 |
| MoCo v2 | 0.58 |
| CLIM | 0.65 |

Table 13: Results of different training epochs

| Epochs | Accuracy (%) |
|--------|--------------|
| 200 | 72.3 |
| 800 | 75.2 |
| 1200 | 75.5 |

**Results of Different Training Epochs**. In Table 13, we compare CLIM trained with different epochs. Our method achieves an accuracy of 72.3% with only 200 epochs, 75.2% with 800 epochs, and can be further improved to 75.5% when training with 1200 epochs.

