# OpenReview forum: "Center-wise Local Image Mixture For Contrastive Representation Learning"
_ICLR.cc/2021/Conference — Reject_

### Official Review · AnonReviewer3 · 2020-10-27
**Interesting but small contribution over sota**

**Rating:** 6
**Confidence:** 4

**Review:**

The paper presents an improved positive sample selection and data augmentation method for unsupervised, contrastive representation learning. Authors propose two improvements to be used in contrastive representation learning: a positive sample selection scheme (called center-wise sample selection), that improves over previously proposed kNN or k-means methods; and multi-resolution data augmentation which is an extension of a known crop augmentation technique with multiple scales.

Strengths:
The impact of the proposed improvements was thoroughly evaluated on several unsupervised feature learning benchmarks. It was shown that the proposed improvements advance state-of-the-art. Results on linear classification using features learned in unsupervised way; in few-shot learning task (using 1% of available labels) on  ImageNet and in transfer learning on VOC object challenge show small but consistent improvement over state-of-the-art.
A large number of recently proposed unsupervised representation learning approaches was included in the comparison. Ablation study proves positive impact of each of the two proposed improvements on the final performance.

Weaknesss:
The proposed improvements show rather limited improvement over state-of-the-art. E.g. from 75.3 (SwAV) to 75.5 (proposed method) on linear classification on ResNet using features learned in unsupervised way; 82.6 (SwAV) to 82.8 (proposed method) in transfer learning on VOC object chanllenge.

Wording and writing style needs to be revised as some sentences are difficult to understand.


In introduction authors write:
"....instance discrimination (Wu et al., 2018) based methods are rapidly closing the performance
gap comparing with the supervised counterparts" and in the following sentence:
"Following this paradigm, self-supervised models are able to generate features that are comparable or even better than those produced by supervised pretraining."
From the first sentence it sounds like unsupervised features still fall behind the features learned in a supervised way. The second sentence claims, that self-supervised models can be even better than those produces by supervised methods. But there's no reference to prove this claim.

"semantic similar images" sounds wrong. Rather it should be "semantically similar images".

"Contrastive learning targets at learning an encoder that is able to map positive pairs to similar representations
while push away those negative samples in the embedding space." does not sound right, especially usage of word "those", rephrase.

".... while current contrastive strategy does not consider the semantic similarities among different samples,
samples, which makes the optimization contradictory and hard for convergence. To solve this issue, we propose a new
kind of data augmentation, termed as Center-wise Local Image Mixture...."
This is an overstatement. It sounds, like there's a serious problem with previous contrastive representation learning approaches and the proposed methods solves them. Whereas, experimental results prove, that the proposed method is only a small improvement over the state-of-the-art.

---

> ### Author Response · Authors · 2020-11-19
> **Response**
>
> We thank the reviewer for the detailed comments. Below, we provide detailed responses to each concern and question.
>
> **Q1: [Marginal improvements over 75.3 (SwAV), and Marginal improvements over VOC [82.6 (SwAV) to 82.8] ]**
>
> CLIM and SwAV are optimized with different objectives, in SwAV, instead of pulling an image with different augmentations together as MoCo, it enforces consistency regularization between cluster assignments of different augmentations. While CLIM proposed a new data augmentation strategy that pulls semantically similar images together, and can be seamlessly incorporated into current contrastive learning based methods to further improve the results. In future work,  we will try to combine CLIM with more works, such as SwAV, BYOL, which we believe the performance will be further improved.
>
> For performance gain, since the supervised baseline is around 76.5, which can be treated as an upper bound of unsupervised learning under this setting, and the improvement is relatively marginal. To better understand the advantages of CLIM over SwAV, we evaluate our representation with KNN classifier, since it does not need to train an additional classifier with labeled data, and is able to evaluate the pretrained features more directly. We center crop the images to obtain features from the last average pooling layer, and implement KNN by faiss. The results are shown in the table below,  CLIM consistently outperforms SwAV in both 20-NN and 200-NN by a large margin.  When fine-tuning with few labels, CLIM also validates its superior performance.
>
> For transferring performance on PASCAL VOC, the limited gain partially results from the high baseline (82.6)  and the domain gap between ImageNet and PASCAL VOC dataset. Most importantly,  we think contrastive learning based methods is not optimal for tasks like detection and segmentation, since detection and segmentation require localization sensitivity, while contrastive learning based methods compress the whole image into a global vector, without considering the spatial information. How to design a pretext task that is suitable for tasks like detection and segmentation remains a future research direction.
>
> | Method      | 20-NN | 200-NN | 1% labels ; Top-1 / Top-5 |
> | ----------- | :---: | :----: | :---------: |
> | Supervised | 75.0  |  73.2  | - |
> | MoCo v2    | 60.1  |  57.2  | 52.4 / 78.4 |
> | SwAV       | 64.7  |  61.0  | 53.9 / 78.5 |
> | CLIM       | 69.6  |  66.7  | 59.3 / 81.6 |
>
> **Q2: [Wording and writing style needs to be revised]**
>
> Thanks for your detailed comments, we have carefully proofread the paper in the revised version.

---

### Official Review · AnonReviewer2 · 2020-10-27
**new data augmentations for improved contrastive learning; questions on empirical validation**

**Rating:** 6
**Confidence:** 4

**Review:**

This paper focuses on contrastive learning for performing self-supervised network pre-training.  Two components are proposed: First, to select semantically similar images that are pulled together in the contrastive learning, the paper proposes "center-wise local image mixture" (CLIM) - both k-means clustering and knn neighbors are computed, and then for a given anchor image x, and images x' that fall within the same cluster and are a knn neighbor (and additionally closer to the cluster center than x) are selected as a positive match to x.  This is motivated from the perspective of allowing for consideration of both local similarity and global aggregation.  This selection is further modified by the use of cutmix data augmentation, where (x, x') are combined via a binary mask.  This is motivated from the perspective of allowing for some smoothing regularization to handle potentially noisy matches from CLIM.

The second component is multi-resolution data augmentation, a variant of crop augmentation that focuses specifically on enabling scale invariance by maintaining the same aspect ratio and performing the cutmix augmentation at different image resolutions.

Positives:
+ interesting proposed method for expanding the neighborhood space of considered positive matches for contrastive learning
+ ablation study provided to show the improvement from each proposed component (sample selection, cutmix, multi-resolution)
+ generally good empirical performance on several tasks - linear evaluation on imagenet, semi-supervised learning with few labels, transfer learning

Neutral:
- overall novelty is moderate; I would consider the main novelty to be in the selection of positive matches, as the cutmix and multi-resolution augmentations are largely leveraging existing ideas.

Negatives:
- ablation studies only use 200 training epochs.  From appendix C, is it clear there is a big difference in accuracy from 200 to 800 or more epochs.  I would like to know how the improvements from the proposed methods still hold up after longer training.
- hyper-parameter selection: there are several different parameters to be set in the proposed work: multi-resolution scales, number of clusters and neighbors for CLIM, alpha in cut-mix, with the differences in accuracy between settings approaching the difference between say knn+cutmix and center-wise+cutmix.  It appears that these hyper-parameters were directly set using the ImageNet linear evaluation, which seems like it has some potentially for overfitting then.  I would have liked to know how well the results generalize if these hyper-parameters are set on some separation validation data.
- as a more minor point, I'm curious how much the restriction in equation (2) matters for CLIM - in other words, what if instead of equation (2), simply let $\Omega_p = \Omega_1 \cap \Omega_2$?

Overall summary:

Given the overall improvements from the proposed method, I'd be inclined toward accept, if the concerns I raised regarding the empirical evaluation were addressed.

---

> ### Author Response · Authors · 2020-11-19
> **Response**
>
> We thank the reviewer for the detailed comments. Below, we provide detailed responses to each concern and question.
>
> **Q1: [how the improvements from the proposed methods still hold up after longer training]**
>
> Thanks for the suggestion. Due to the limited time, we compare two ablation results with our method, as well as moco v2 baseline with 800 epochs. We would add more comparisons in the revised version.
>
> | Method (800 epochs)               | Linear eval; Top-1 |
> | --------------------------------- | :----------------: |
> | MoCo v2                           |        71.1        |
> | Center-wise + cutmix              |        73.7        |
> | Center-wise + cutmix + Multi-reso |        75.2        |
>
>
>
> **Q2: [hyper-parameter selection]**
>
> A good question. We agree it is not rigorous to tune the parameters and report results on the same validation set, although most previous works like Moco, SimCLR follow this settings.  However, the advantages of the proposed center-wise samples selection strategy can also be validated by fine-tuning with few labels, where we do not tune parameters on this settings. As shown in the table below,  center-wise method brings about 4.4% gain when fine-tuning with 1% labels, which demonstrates the advantages of center-wise sample selection strategy.
>
> | Method (200 epochs) | 1% labels ; Top-1 / Top-5 |
> | ------------------- | :-----------------------: |
> | knn                 |        45.1 / 69.1        |
> | center-wise         |        49.5 / 75.3        |
>
> To better validate the advantages of the proposed CLIM method, we divide the ImageNet training dataset into two splits, shown in the table below, and train the model with the new train split from scratch and validate the performance on the new validation set, using the best tuned parameters in the paper, and the results are shown in the table below. It can be shown that center-wise selection strategy consistently outperforms the knn method.
>
> | Dataset  | Original | Training samples | Valid samples |
> | -------- | -------- | ---------------- | ------------- |
> | ImageNet | 1281167  | 1231167          | 50000         |
>
> | Method (200 epochs) | w/o cutmix | w/ cutmix |
> | ------------------- | :--------: | :-------: |
> | knn                 |    68.0    |   69.4    |
> | center-wise         |    69.1    |   69.9    |
>
> **Q3: [Minor Point: Abalation study on sampling of positives from the top 40 nearest neighbors within the cluster]**
>
> Thanks for your suggestion, we add an experiment for $\Omega = \Omega_1 \cap \Omega_2 $, following the best setting in Table 8 (m=10,000 k=40), and report the results in the table below. Simply selecting the union achieves 69.6% and 68.5% accuracies with and without cutmix, respectively, which is comparable with results that directly using knn. Since for samples not lie around the boundary, it equals to knn, and does not encourage intra-class compactness as CLIM. We have added this ablation study in the revised version.
>
>
> | Method (without Multi-Reso)        | w/o cutmix | w/ cutmix |
> | ---------------------------------- | :--------: | --------- |
> | $\Omega = \Omega_1 \cap \Omega_2 $ |    68.5    | 69.6      |
> | Center-wise                        |    69.3    | 70.1      |

---

### Official Review · AnonReviewer4 · 2020-10-30
**Simple method, very strong results, but not very novel and some justifications are missing.**

**Rating:** 6
**Confidence:** 4

**Review:**

Center-wise Local Image Mixture For Contrastive Representation Learning

The paper introduces a new contrastive learning method for unsupervised representation learning. The main idea is to consider the semantic similarity between different images and incorporate it in the learning procedure, in contrast to the many contrastive learning methods which only used augmentations of the query image as positives.
The main contribution is 2-fold: a) use nearest neighbors from the same cluster which are closer to the centroid than the anchor as positive samples; b) use more complex augmentations, i.e. [CutMix] and multi-resolution during training. The proposed method achieves state-of-the-art results for unsupervised learning on Imagenet and transfer learning tasks on Pascal VOC, COCO, and LVIS.

### Pros

+ The paper is written well and easy to understand.
+ The method is simple and yet powerful, achieving state-of-the-art results on several standard benchmarks.
+ The method is easy to implement and reproduce.
+ The paper shows the importance of better modeling of intra-class variance by means of sampling positives among the nearest neighbors.
+ I appreciate quite detailed ablation studies (but not all of them).

### Cons
- The main ideas presented in the paper are not entirely new.
  - The idea of using highly related images in the learned so far representation space as positives (vs single exemplar + its augmentations or just naive nearest neighbors) for unsupervised representation learning was already explored in [CliqueCNN].
  Since the proposed positive sampling method is the cornerstone contribution of this paper, it would be nice to see how it compares with the sampling method proposed in [CliqueCNN], where only the nearest samples which form a clique (mutually very similar samples) are used as positives.
  - [CutMix] augmentation is a previously published work. And the contribution of this paper is applying CutMix in the context of contrastive learning, which is yet another augmentation among a huge variety of possibilities. E.g., CutOut, MixUp, Attentive CutMix, etc. The paper gives no justification for why CutMix is especially better in this context.
  - Similar Multi-resolution augmentation and the importance of using different image resolutions during raining and testing were explored in [FixRes].

- p.4 *"Cluster-based method regards all samples that belong to the same center as positive pairs, which breaks the local similarity among samples especially when the anchor is around the boundary."* It is not very clear how the proposed method differs from the clustering-based methods in this sense. It seems like the proposed sampling method can also break local similarity around the cluster boundary because the anchor will be attracted to the cluster center increasing the distance to the samples from other clusters, which will result in very pronounced hubs in the representation space (as in Fig. 3).
- It is not clear from the experiments what is the main performance booster compared to the KNN baseline in Tab. 6. Is it (A) the sampling of positives from the top 40 nearest neighbors within the cluster (as explained in A.2) or (B) discarding the positive which are further from the cluster centroid than the anchor?  The experiment in Sec. 4.4 do not answer this question, because the K-means baseline in Tab.6 randomly selects positive samples from the entire cluster and not from the top 40 closest samples within the cluster. To prove that (B) is crucial one would need to make an extra ablation study (A), where the positives are sampled from the top 40 nearest neighbors within the cluster (w/o discarding those which are further from the centroid than the anchor).
- *Minor*. Would be nice to see and extra ablation experiment on the full training schedule (1200 epochs) where all paper contributions are enabled one by one, e.g.: (i)  baseline, (ii) baseline + proposed positive sampling, (iii) baseline + proposed positive sampling + CutMix, (iv) baseline + proposed positive sampling + CutMix + multi-res.
I understand that it is computationally demanding, however, it would provide a better picture of the performance contribution of the final components (after tuning them using a shorter training schedule with 200 epochs), since improvements brought by some components on the shorter training schedule can become less significant when the network is trained longer.


## After rebuttal
After reading the authors comments' and other reviews, I think that this is a **borderline** paper that could benefit from more rigorous experimental validation.

[CutMix] CutMix: Regularization Strategy to Train Strong Classifierswith Localizable Features, ICCV 2019.
[CliqueCNN] CliqueCnn: Deep unsupervised exemplar learning, Bautista et al., NeurIPS 2016.
[CutOut] Terrance DeVries and Graham W Taylor. Improved regularization of convolutional neural networks with cutout, 2017.
[MixUp] Mixup: Beyond empirical risk minimization, Zhang et al., 2017.
[Attentive CutMix]  Attentive CutMix: An Enhanced Data AugmentationApproach for Deep Learning Based ImageClassification, Walawalkar et al., 2020.
[FixRes] Fixing the train-test resolution discrepancy, Touvron et al., NeurIPS 2019.

---

> ### Author Response · Authors · 2020-11-19
> **Response (Part 1/2)**
>
> We thank the reviewer for the detailed comments. Below, we provide detailed responses to each concern and question.
>
> **Q1: [Compare with CliqueCNN]:**
>
> These two methods are different in defining similar samples. In clique cnn, samples that are mutually similar are grouped into a clique. This is achieved by starting with few samples that are mutually similar, and use farthest neighbor clustering to merge cliques in a bottom up way. However, merging cliques would inevitably introduce samples that are relatively far away into a meta-clique. Different from clique cnn, for an anchor image, we simply select knn samples that are closer to the corresponding center, and pull the anchor closer with these samples. In this way, the samples are pulling towards the center in a progressive way, while do not destroy the local similarity. **Note that we do not require compactness among the selected samples of an anchor, and also do not explicitly pull these samples together**. As a result, the pulled samples in CLIM are much more compact comparing with similar samples defined clique cnn.
>
> **Q2: [CutMix is a previously published work, why CutMix is better in this context]:**
>
> In fully supervised learning, CutMix randomly selects two images for mixing, and changes the labels accordingly. This cannot be directly transferred for unsupervised learning, as shown in Table 6, randomly mixing two images degenerates the performance. To solve this issue, we **conduct CutMix only within the local neighborhood of an image** (with deliberately designed center wise positive sample strategy), which does not break the local similarity when pulling these mixed samples. This is not trivial, and the mixing augmentation can be seamlessly integrated into current contrastive learning based methods for further performance gain.
>
> Since our method targets at generating new samples that expanding the neighborhood of an anchor, mixing methods such as mixup also holds, but may be different in performance gain.  As shown in the table below, we implement mixup with our proposed method. Here we try different choices of $\alpha$ in beta distribution for mixup and choose the best one ($\alpha$ = 0.2) for comparison. Cutmix is better than mixup, partially because mixup destroys the real pixel  distribution(destroys the naturality of pixels). However, cutout only cuts a patch out of the image, which cannot behave like a data mixing operation. While Attentive cutmix needs a pretrained classification network to generate attention feature map, which is also not applicable for unsupervised learning. We have add these comparisons in the revised version.
>
> | Methods | Mixup (200 epochs; Linear Eval) | Cutmix (200 epochs; Linear Eval) |
> | :-----: | :-----------------------------: | :------------------------------: |
> |  CLIM   |              69.5               |               70.1               |
>
> **Q3: [Compare multi-res augmentation with fixRes ]:**
>
> Both methods explore resolution issue, but are different in both motivation and goal. FixRes is based on the observation that data augmentations induce a significant discrepancy between the size of the objects seen by the classifier at train and test time, and employs different train and test resolutions to fix the train-test resolution discrepancy. **The goal is to require less scale invariance for the neural net in FixRes.** **While our multi-resolution augmentation aims to model the scale invariance explicitly**, which is not carefully considered in previous self-supervised learning. We have added the comparison in the revised version.
>
> **Q4: [Not very clear how the proposed method differs from the clustering-based methods ]**
>
> From Fig. 2, the clustering-based method regards all samples that belong to the same cluster as positive pairs, and would **explicitly pull those samples that are dissimilar together**, this is especially true when the anchor is around the cluster boundary. While center-wise sample selection constrains similar samples to be selected in the local neighborhood of an anchor, and thus ensure the local similarity.  For boundary samples that are from two different clusters, **these samples are dynamically changing as the training process goes, and we do not conduct any constraint among them.**

---

> > ### Comment · AnonReviewer4 · 2020-11-24
> > **Do not agree with  the answer**
> >
> > I want to thank the authors for their detailed response. However, I do not agree with the answer to Q1.
> >
> > > "[CliqueCNN]... uses the farthest neighbor clustering to merge cliques in a bottom-up way. However, merging cliques would inevitably introduce samples that are relatively far away into a meta-clique"
> >
> > I respectfully disagree with this statement. Merging cliques would not combine samples that are relatively far away from each other, because in the farthest neighbor clustering the linkage function is the maximal distance between points within two cliques. Hence, one can easily prevent two cliques from being merged if the linkage function value is higher than some threshold.
> >
> > > "we do not require compactness among the selected samples of an anchor"
> >
> > On the contrary, the fact that the proposed method does not require compactness among the selected positive samples means that it can pull together very unrelated images which is also shown in [CliqueCNN].
> >
> > > "the pulled samples in CLIM are much more compact comparing with similar samples defined clique cnn."
> >
> > This sounds a little bit hand-wavy and would require experimental or theoretical grounding.
> >
> > ---
> >
> > Overall, I think the paper would benefit from a more thorough analysis of the positive sampling since the proposed method is very similar in mechanics to the self-supervised method [CliqueCNN], which also considers the semantic similarities among different samples, and not only different views of the same sample for learning positive relationships. Moreover, some statements contradict the findings described in [CliqueCNN] (i.e just taking arbitrary NNs as positives is usually not sufficient as the not converged feature representation used for the retrieval is not very confident which would result in NNs being too diverse).

---

> > > ### Author Response · Authors · 2020-11-25
> > > **The positive sample selection in clique CNN  is a widely used unsupervised clustering algorithm ,  and not the main contribution of clique CNN**
> > >
> > >  **Q: Merging cliques would not combine samples that are relatively far away from each other**
> > >
> > > Sorry for the misleading, here **we claim 'far away' is compared with our center-wise sample selection strategy**. In clique CNN, the samples are grouped via complete-linkage clustering starting with few mutually similar samples.  Although farthest-neighbor clustering ensures that the clustered samples are more compact compared with standard k-means, **it is sensitive to outliers and may produce unreasonable clusters.** In clique CNN,  the similarity computation is based on the inferred object bounding boxes to reduce noise influence. **Notably, Clique CNN is based on mutually similarity, while in our method it is only based on distance to a given anchor**, and we do not enforce pairwise similarity among the selected positive samples for an anchor. In clique CNN, the two samples that are most similar may not be grouped in order to ensure mutually similarity for all samples.  As a result, for the same number of similar samples, the average distance of the positive samples to the anchor is nearer comparing with the average distance of pairwise similarity in clique CNN.
> > >
> > > **Q: it can pull together very unrelated images**
> > >
> > > We pull together samples that are similar to the anchor,  but do not **explicitly** pull the positive samples that belong to neighbors of the same anchor, since these samples may not be mutually similar. The distance among these samples are evolved via the generalization of the learned representation. This would  not  result in pulling together very unrelated images.
> > >
> > >
> > > **Q: the proposed method is very similar in mechanics to the self-supervised method [CliqueCNN]**
> > >
> > > After carefully reading clique CNN, we claim that **the positive sample selection in clique CNN  is de facto a widely used unsupervised clustering algorithm that based on complete-linkage similarity criterion and agglomerative hierarchical clustering method,  which is time consuming comparing with standard k-means and** **this is not the main contribution of clique CNN**.  Instead the contribution of clique CNN is  to formulate an optimization problem that builds training batches for the CNN by selecting groups of compact cliques, so that all cliques in a batch are mutually distant. This is totally different from our method that proposes a center-wise sample selection strategy, and apply data mixing augmentation to introduce new samples that lie around the local neighborhood of an anchor, but current representation cannot model well.  **The compactness in feature representation is ensured by pulling center-wise neighbors corresponding to an anchor, but not via deliberately selected samples that are mutually similar and pulling them together**. We find that selecting mutually similar samples are much more difficult than our center-wise selection strategy.  Furthermore,  agglomerative clustering method is time consuming, which is not suitable for large scale ImageNet dataset, and clique CNN only validates it on small scale dataset like Olympic Sports dataset and pascal voc. While we are able to efficiently (less than 5min) update the positive samples using standard k-means and knn algorithms.
> > >
> > > **Q: some statements contradict the findings described in CliqueCNN**
> > >
> > > The features used in clique CNN is initially based on low-level HOG-LDA features, and the representation ability is much lower than that based on Moco baseline. Hence for a range of k in knn, the probability of selecting correct samples are higher than that in clique CNN. In Table 6, we also show that randomly selected positive samples would deteriorate the performance (67.5->62.3) without data mixing.

---

> ### Author Response · Authors · 2020-11-19
> **Response (Part 2/2)**
>
>
> **Q5: [Abalation study on sampling of positives from the top 40 nearest neighbors within the cluster]**
>
> Thanks for your suggestion, we add an experiment on this setting and report the results in the table below. Using top-40 knn within the cluster achieves 69.6% and 68.5% accuracies with and without cutmix, respectively, which is comparable with results that directly using knn. Since for samples not lie around the boundary, it equals to knn, and does not encourage global aggregation as CLIM. We have added this ablation study in the revised version.
>
> | Method (200 epochs)                         | w/o cutmix | w/ cutmix |
> | ------------------------------------------- | :--------: | :-------: |
> | top 40 nearest neighbors within the cluster |    68.5    |   69.6    |
> | Center-wise                                 |    69.3    |   70.1    |
>
> **Q6: [Minor: extra ablation experiments on the full training schedule]**
>
> Thanks for your suggestion. As you know, it is computationally demanding to report ablation studies for 1200 epochs. Due to the limited time, we compare partial results for 800 epochs, shown in the table below.
>
> | Method (800 epochs)               | Linear eval; Top-1 |
> | --------------------------------- | :----------------: |
> | MoCo v2                           |        71.1        |
> | Center-wise + cutmix              |        73.7        |
> | Center-wise + cutmix + Multi-reso |        75.2        |

---

### Official Review · AnonReviewer1 · 2020-11-02
**Would like to see a more extensive explanation of the motivation behind the method**

**Rating:** 5
**Confidence:** 4

**Review:**

## Summary

The paper addresses the problem of contrastive representation learning, and proposes a new data augmentation, dubbed CLIM, that leverages similarity between images. Instead of generating positives pairs using different transformation of the same image -as it is standard in contrastive learning-, positive pairs are generated using those similar images to the anchor image: after clustering the representation space using k-means, the nearest neighbours that are closer to the corresponding center of the cluster where the anchor belongs to are selected. Then, positive pairs are constructed following Cutmix, which can be seen as a regulariser, and which consists in cutting and pasting patches among these pairs to generate new samples. Finally, the paper also proposes a multi-resolution augmentation, which consists in random zooms in (ie. random crop + resize) at different scales to enable scale invariance.


## Pros

The paper proposes a simple, yet effective, data augmentation method that seems to help learning stronger representations for some tasks.

The experimental section is good, which includes evaluations for different tasks and an ablation study that analyses the contribution of the different components of the proposed approach.


## Cons

I didn’t find that the proposed approach was properly motivated in the paper. After reading the paper I didn’t get a good understanding of why ensuring that the images are pulled towards the center of these clusters is a good property for learning good representations. Why is it better to pull samples towards the center instead of simply pulling them towards the nearest neighbours? I missed a more theoretical and extensive explanation about this point, which is basically the core idea of the proposed approach. The authors motivate this by saying that “it is better to encourage global aggregation property while pulling local similar samples”, which I couldn’t understand. I think the paper would benefit from an extended motivation where the authors could elaborate more on this.

The presentation and the writing could be improved.

While the improvements that the CLIM augmentation brings seem to be quite good in the linear evaluation on ImageNet by boosting the results of MoCo v2 (which is used as a baseline if I understood correctly) by 4 points, the improvement of this representation on other downstream tasks (ie. detection on PASCAL, COCO, LVIS) is minimal. Results are on par with MoCo v2 in all the scenarios (+0.4% in the best case), which means that the CLIM augmentation is not really bringing a significant improvement to the representation in these cases.

Related to the point above, the authors claim in the experiments on LVIS that “CLIM outperforms […] MoCo v2 by a large margin”. I have to disagree with this statement, since I personally consider these results to be on par: it only outperforms MoCo by 0.4/0.3 points in AP. I would suggest the authors to rephrase this claim.

Is MoCo v2 used as a baseline setting where CLIM augmentations are applied to? I understood this from the explanation in sections 4.1 and 4.3, but this is not clear for the semi-supervised training experiments in Section 4.2. If so, then what is the performance of MoCo v2 in semi-supervised training on ImageNet (table 2)? It would have been interesting to see what’s the improvement that CLIM brings rather than a direct comparison with other methods.


## Recommendation

My initial recommendation is leaning towards reject. Although the augmentation proposed is simple (in the good sense) and seems to work well (in some cases), I found that the paper somehow failed at motivating it, which I think it’s important for a paper like this. I also think results are a bit underwhelming for downstream tasks since they are on par with the baseline.

---

> ### Author Response · Authors · 2020-11-19
> **Response**
>
> We thank the reviewer for the detailed comments. Below, we provide detailed responses to each concern and question.
>
> **Q 1: [Motivation for pullings images towards the center of clusters]:**
>
> The motivation is based on the fact that a good representation should be endowed with high intra-class similarity. We analyze the intra-class similarity (r.f. Table 10 ) of moco at different epochs, and find that although moco does not explicitly model invariance to similar images, the intra-class similarity becomes higher as the training process goes. Based on this observation, we explicitly enforce similar images towards the center of clusters, and generate representation with higher intra-class similarity, which we find  is beneficial for semi-supervised learning (especially with 1% labels). While knn simply models similarity among local neighborhood and does not enforce global compactness. Comparing with CLIM, it suffers low intra-class similarity and achieve inferior performance when fine-tuned with 1% labels. We would clarify this in the revised version.
> _______________________________________________________________
> | Methods                          |    Intra-Class Similarity    |    1% labels ;  Top-1 / Top-5    |
> | :--------------------- | :--------------------: | :------------------------: |
> | MoCo V2 200 epochs     |          0.54          |         42.3/67.7          |
> | MoCo V2 800 epochs     |          0.58          |         52.4/78.4          |
> | KNN 200 epochs         |          0.55          |         45.1/69.1          |
> | Center-wise 200 epochs |          0.57          |         49.5/75.3          |
> ________________________________________________________________
>
> **Q2: [The presentation and the writing could be improved]:**
>
> Thanks for the suggestion, we have carefully proofread the paper in the revised version.
>
> **Q3: [Limited performance gain on downstream tasks PASCAL VOC and COCO]：**
>
> A good question.  We do not claim significant improvement on these two tasks, and in the paper we state"slightly better than MoCo v2". We think that this is the bottleneck of current instance discrimination based methods for transferring to detection and segmentation tasks. Since detection and segmentation requires localization sensitivity, while contrastive learning based methods compress the whole image into a global vector, without considering the spatial information. The improvement for classification does not directly correspond to improvement on the two tasks, and we find that recent methods such as SimCLR, SwAV and BYOL all outperforms MoCo v2, but all brings about marginal improvement or even lower performance comparing with MoCo v2. How to design a pretext task that is suitable for tasks like detection and segmentation remains a future research direction.
>
> **Q4: [Not a large margin improvement comparing with MoCo v2 on LVIS]:**
>
> Thanks for pointing out this, we have revised this expression accordingly.
>
> **Q5: [Comparision with MoCo V2 on semi-supervised setting]:**
>
> Yes, CLIM uses MoCo v2 as baseline for model pre-training, and we follow the same setting as in MoCo v2 for linear evaluation in Section 4.1, and downstream tasks in section 4.2.  Since **MoCo v2 does not report results under semi-supervised settings**, we follow the setting as in SimCLR , and all results in Table 2 are from the original paper. We also evaluate the performance of MoCo v2 with the provided model of 800 epochs, using the same setting for fair comparisons. The results are shown in the table below. CLIM consistently outperforms MoCo v2 by a large margin, and we have added the results in the revised version.
> ________________________________________________________________
> | Methods       |    1% labels (Top-1 / Top-5)    |   10% labels (Top-1 / Top-5)   |
> | :------ | :-----------------------: | :------------------------: |
> | MoCo V2                        |        52.4 / 78.4        |        65.3 / 86.6         |
> | CLIM                           |      **59.3 / 81.6**      |      **70.0 / 89.3**       |
> ________________________________________________________________

---

### Author Response · Authors · 2020-11-23
**General Response**

We thank all reviewers for the thoughtful feedback. We have addressed the concerns from the reviewers below and provided the suggested modifications in the revised manuscript and highlight them in red.

---

### Decision · Program_Chairs · 2021-01-07
**Final Decision**

**Decision:**

Reject

**Comment:**

There are two main contributions in this paper. First, the use of NN from the same cluster as “views” of the data as understood in classical contrastive learning. Second, the use of additional augmentation techniques, namely cutMix and multi-resolution. The reviewers noted that the paper is written well and easy to understand, that the ablation study is conducted well and that the model shows good empirical performance on several tasks.

At the same time, the somewhat limited novelty of the paper was also discussed. As noted by R4, all aspects of the present paper have been discussed in previous work. The difference with previously published clustering-based SSL methods was also not very clear. This was discussed in the rebuttal but without strong evidence supporting the claims. Moreover, the ablation study is conducted on models that are trained for 200 epochs. While this is understandable from a pragmatic point of view, the conclusions may be completely different when the model is fully optimised.

Because of all the points raised in the discussions, this paper is a too close to borderline to be accepted. We recommend the authors improve the manuscript given the feedback provided in the reviews and discussion and resubmit to another venue.